# Tumor Microenvironment, Inflammation, and Inflammatory Prognostic Indices in Diffuse Large B-Cell Lymphomas: A Narrative Review

**DOI:** 10.3390/ijms26125670

**Published:** 2025-06-13

**Authors:** Zorica Cvetković, Olivera Marković, Gligorije Marinković, Snežana Pejić, Vesna Vučić

**Affiliations:** 1Department of Hematology, University Hospital Medical Center Zemun, 11080 Belgrade, Serbia; 2Faculty of Medicine, University of Belgrade, 11000 Belgrade, Serbia; 3Department of Hematology, University Hospital Medical Center Bežanijska kosa, 11080 Belgrade, Serbia; 4Department of Molecular Biology and Endocrinology, Institute of Nuclear Sciences, National Institute of the Republic of Serbia, University of Belgrade, 11000 Belgrade, Serbia; 5Center of Research Excellence in Nutrition and Metabolism, Institute for Medical Research, National Institute of the Republic of Serbia, University of Belgrade, 11000 Belgrade, Serbia

**Keywords:** diffuse large B-cell lymphoma, DLBCL, tumor microenvironment, inflammation, innate immunity, adaptive immunity, prognostic index

## Abstract

Diffuse large B-cell lymphoma (DLBCL) is the most common subtype of non-Hodgkin lymphoma, characterized by significant variability in clinical outcomes. Emerging evidence highlights the pivotal role of inflammation in the pathogenesis and prognosis of DLBCL. This narrative review explores the interplay between the tumor microenvironment, inflammatory processes, and prognostic indices used in DLBCL, focusing on biomarkers, immune responses, and systemic inflammation. These indices show promise as predictive and prognostic tools comparable to molecular markers, such as gene expression profiling, which are currently considered gold standards in prognosis but are often costly and technically demanding. By synthesizing findings from the current literature, this article highlights the potential of inflammatory indices as accessible and cost-effective prognostic alternatives to molecular markers in DLBCL, while also underscoring the need for further research to validate their clinical utility.

## 1. Introduction

Non-Hodgkin lymphomas (NHLs) are the 10th most common malignancy worldwide, according to the most recent GLOBOCAN cancer statistics from the International Agency for Research on Cancer for the year 2020 [1]. Among NHL subtypes, diffuse large B-cell lymphoma (DLBCL) is the most prevalent, accounting for over a third of all NHL cases, with an estimated age-standardized incidence rate of 7.2 per 100,000. In Western countries, the incidence of DLBCL has remained stable since the beginning of the 21st century, likely due to improved control of known host and environmental risk factors, such as obesity, infections, and exposure to toxins [2].

DLBCL, not otherwise specified (NOS), is characterized by an aggressive course and a heterogeneous morphological, molecular, and clinical presentation. Both of the currently coexisting classification systems—the 5th edition of the World Health Organization (WHO) Classification of Hematolymphoid Tumors (WHO-HAEM5, 2022) [3] and the International Consensus Classification (ICC, 2022) [4]—still recommend subtyping according to the cell of origin into DLBCL germinal center B-cell-like (GCB) and DLBCL activated B-cell-like (ABC, non-GCB), with the latter generally associated with a worse prognosis.

The standard first-line treatment for DLBCL is chemoimmunotherapy (CIT), typically combining rituximab with an anthracycline-based chemotherapy regimen such as cyclophosphamide, doxorubicin, vincristine, and prednisone (CHOP) or a CHOP-like alternative. While the majority of patients with DLBCL respond favorably to initial therapy, approximately 40% either fail to achieve remission or experience relapse following treatment [5]. Improved insights into the biology of DLBCL have led to the development of new therapeutic strategies aimed at enhancing patient outcomes and survival rates [6]. However, despite a slight but steady decline in mortality rates in recent years, the 5-year relative survival rate, which remains around 65%, is still considered unsatisfactory [7].

Advancements in modern molecular and genetic techniques have significantly improved our understanding of lymphoma pathogenesis and enabled more accurate prognostic assessments. However, these methods are often costly, time-consuming, and not widely available in many clinical settings. In parallel with the ongoing research into molecular markers as potential therapeutic targets, increasing attention is being given to the development of simple, cost-effective, and accessible clinical prognostic indices. These tools, alongside established models such as the International Prognostic Index (IPI) [8,9], its revised version [10], the age-adjusted IPI [8,11], and the National Comprehensive Cancer Network International Prognostic Index (NCCN-IPI) [12,13], are being investigated to support treatment decisions and follow-up care in patients with DLBCL. The goal of these new indices is to complement existing models by offering additional, clinically practical insights into disease progression, treatment response, and patient outcomes.

Inflammation is a well-established contributor to the development and progression of various cancers, including hematologic malignancies [14]. Emerging therapies, such as immune checkpoint inhibitors (ICPis) targeting cytotoxic T-lymphocyte-associated antigen-4 (CTLA-4), programmed cell death-1 (PD-1) and PD ligand 1 (PD-L1), bispecific antibodies (BsAbs), and chimeric antigen receptor (CAR) T-cell therapy, aim to enhance T-cell-mediated antitumor immunity [15]. In parallel, inflammatory markers and indices are increasingly studied as prognostic tools in oncology. In DLBCL, inflammation plays a key role in shaping the tumor microenvironment (TME) and promoting tumorigenesis through immune cell recruitment, cytokine production, angiogenesis, dysregulated signaling pathways, and altered apoptosis. Inflammation not only drives disease progression but also influences treatment response [6,16,17].

Despite the growing evidence, there remain significant gaps in understanding which specific inflammatory markers and indices most accurately predict outcomes in DLBCL, how they interact with treatment responses, and their integration into routine clinical decision making. This narrative review aims to address these gaps by highlighting the role of lipid- and protein-derived inflammatory markers, peripheral blood and TME immune cells, and inflammatory indices as emerging predictive and prognostic biomarkers in DLBCL, with an emphasis on their clinical utility and potential to guide personalized therapy.

## 2. Lipids and Lipid-Derived Inflammatory Markers in DLBCL

Lipid profile disturbances, commonly associated with inflammatory markers, have been consistently observed in patients with a range of malignancies, including NHL, over the past several decades. Recent studies have highlighted altered lipid metabolism as a major oncogenic factor in aggressive forms of NHL, including DLBCL [18]. Lipids are essential parts of cellular membranes, contributing to their structural integrity and fluidity. They also serve as important secondary messengers in various inflammatory and oncogenic pathways and provide a vital source of energy [19,20,21], as illustrated in Figure 1.

### 2.1. Cholesterol

A large prospective study, tracking 61,936 patients from the IQVIA (formerly Quintiles and IMS Health, Inc., Durham, NC, USA) database between 2005 and 2019, found that individuals with reduced levels of high-density lipoprotein cholesterol (HDL-C) had a significantly higher risk of developing multiple types of cancer, including hematological malignancies [22]. Notably, a significant and prolonged decrease in HDL-C levels was detected up to 10 years prior to the clinical onset of DLBCL and other NHL subtypes when compared to healthy individuals [23,24,25]. The reduction in cholesterol and HDL-C levels in the plasma of NHL patients correlates with the aggressiveness and clinical stage of the disease, while a sustained increase back to normal levels has been observed in NHL patients who achieved remission [26,27,28]. Furthermore, HDL-C showed remarkably predictive value for progression-free survival (PFS) and overall survival (OS) in patients treated with CIT [29]. Gao et al. combined NCCN-IPI and the cholesterol status to develop a novel prognostic score, Lipo-PI, which demonstrated superior accuracy over NCCN-IPI in predicting 3-year and 5-year OS and PFS [30].

The major structural protein of HDL-C is apolipoprotein A (apoA), which serves as an activator of lecithin: cholesterol acyltransferase (LCAT), an enzyme that converts free cholesterol into cholesteryl ester. In a large retrospective study on 949 newly diagnosed DLBCL patients, lower levels of apoA1 correlated with lymphoma burden symptoms, advanced stage, increased rate of relapse, higher incidence of refractory disease, and shorter OS [31]. Similarly, in a retrospective analysis of 105 DLBCL patients, Yu et al. demonstrated that apoA1 was an independent predictor of OS and PFS and introduced a novel IPI-A score that combines IPI with apoA1 levels [28]. These findings suggest that lipid profiles may serve as non-specific prognostic and predictive biomarkers for DLBCL, reflecting tumor proliferative capacity, tumor burden, and potentially treatment response.

As a crucial component of the cell membrane in both malignant and normal cells, cholesterol is essential for maintaining membrane integrity and fluidity [32]. The cause of cancer-related hypocholesterolemia is more complex than simply the enhanced receptor-mediated influx and consumption of cholesterol by rapidly proliferating tumor cells [19].

It is well established that dyslipidemia, chronic inflammation, and oxidative stress interact synergistically to drive tumorigenesis. The beneficial, multifaceted protective effect of HDL-C in mitigating chronic inflammation states, including in solid and hematological cancers, is still under extensive investigation. Cholesterol regulates numerous processes, including ferroptosis, autophagy, the DNA damage response, the expression of ICP genes, and influences the TME. Specifically, cholesterol modulates the immune response of TME cells through several mechanisms: inhibiting antigen presentation, enriching immunosuppressive cells like neutrophils and tumor-associated macrophages (TAMs), and regulating immune-effector cells such as CD8^+^ T-cells and natural killer (NK) cells [19,33,34]. These effects occur via both contact-dependent and contact-independent mechanisms, involving soluble molecules like growth factors and lipid-derived cytokines [35,36]. In a recently published prospective observational study by Ou W et al., HDL-C was proposed to be a cardioprotective target, based on the finding that patients with DLBL treated with the standard R-CHOP regimen and higher HDL-C levels exhibited a significantly lower rate of anthracycline-induced subclinical cardiotoxicity [37].

### 2.2. Plasma Phospolipids

In the context of NHL and plasma phospholipid (PL) fractions, our previous research observed a baseline reduction in all plasma PL species—specifically phosphatidylcholine (PC), lysophosphatidylcholine (LPC), phosphatidylethanolamine (PE), and sphingomyelin (SM). These reductions were more pronounced in aggressive NHL subtypes and remained low in patients who progressed on front-line treatment. In contrast, lipid levels tended to increase in patients who responded well to treatment [38]. Regarding the distribution of plasma PL fatty acids (FAs), we found a reduction in polyunsaturated FA (PUFAs) that correlated with both the aggressiveness of NHL [39] and treatment response [40]. Additionally, our recent pilot study using advanced high-resolution ion mobility mass spectrometry on treatment-naïve DLBCL revealed a decline in 54 plasma PL and sphingolipid (SL) species. Most of these species contained linoleic acid (LA; 18:2(n-6)) or arachidonic acid (AA; 20:4(n-6)), which are key precursors of various cytokines [41].

The observed decline in PL levels, along with the altered FA profile, is thought to result from the complex interplay between the uptake of exogenous fatty acids, de novo lipogenesis, and fatty acid oxidation (FAO or β-oxidation) for ATP production [42,43]. This metabolic reprogramming plays a significant role in the pathogenesis of lymphoma.

### 2.3. FA

Cancer cells undergo a decisive transformation in their metabolism, compellingly shifting their energy source from glucose to the more potent FA. The overexpression of fatty acid synthase (FASN), a key enzyme involved in the synthesis of 16-carbon FA palmitate and overall lipid biosynthesis, is associated with poor DLBCL prognosis [44], while its selective inhibition disrupts c-Met receptor signaling and induces apoptosis in DLBCL cells [45,46].

The uptake of exogenous fatty acids is regulated by transmembrane proteins, particularly CD36 and CD37.

CD37, a member of the tetraspanin family almost exclusively expressed in lymphoid tissues, modulates lipid metabolism by inhibiting the fatty acid transporter FATP1. This inhibition restricts the uptake and conversion of exogenous palmitate into energy, its incorporation into structural and oncogenic signaling lipids, and its use in posttranslational modifications such as the palmitoylation of PD-L1 [47,48]. CD37 is significantly downregulated in aggressive B-cell lymphoma cell lines resistant to rituximab; nevertheless, this reduction in CD37 expression does not diminish the therapeutic efficacy of CD37-targeted CAR T-cell therapy [49]. Recognizing the prognostic significance of CD37, Xu-Monette et al. proposed two refined risk stratification models for DLBCL: the molecularly adjusted IPI for R-CHOP (M-IPI-R), which incorporates CD37 negativity and the ABC subtype with the IPI; and IPI + IHC, which integrates immunohistochemical (IHC) expression of CD37, Myc, and Bcl-2 with IPI. Both models significantly enhanced prognostic precision compared to IPI alone. These combined indices demonstrated enhanced prognostic accuracy compared to the IPI alone [50].

CD36, a type 2 cell surface scavenger receptor with three distinct domains, binds to proteins containing thrombospondin structural homology repeat (TSR) domains, molecules exhibiting molecular structures consistent with danger-associated molecular patterns (DAMPs) and pathogen-associated molecular patterns (PAMPs), and functions as a transmembrane channel for exogenous long-chain FA acid uptake and lipid accumulation. CD36 is expressed on the surface of B cells, as well as various TME cells, including macrophages, monocytes, dendritic cells (DCs), and subsets of T cells [51]. Its upregulation has been linked to several key processes in cancer biology, including tumor cell proliferation, immune evasion, and metastasis [52]. In the context of DLBCL, higher levels of CD36 expression on B cells are strongly correlated with the polarization of TAMs toward the M2 subtype, particularly in CD5+ DLBCL [53]. Additionally, elevated CD36 expression is indicative of poorer prognoses for DLBCL patients [54]. This suggests that CD36 and CD37 may play a critical role in shaping the immune landscape of DLBCL and could serve as potential biomarkers for disease progression and response to treatment and appealing targets for novel therapeutic strategies in lymphoma [20,55,56,57].

Dysregulation of AA metabolism can lead to increased inflammatory responses, tumor progression, and immune suppression within the TME. The dysregulation of AA metabolism, a key n-6 PUFA, can lead to increased inflammatory responses, tumor progression, and immune suppression within the TME. The accumulation of AA in the PLs of cell membranes is associated with the ability of inflammatory cells to produce bioactive lipid mediators. These mediators include prostaglandins (PGs), leukotrienes (LTs), epoxyeicosatetraenoic acids (EETs), and endocannabinoids (ECs), all of which play a role in regulating immune cell function and modulating the TME [58,59,60]. AA is released from PLs in the cell membrane by cytosolic phospholipase A2 (cPLA2) and further catabolized through various pathways, primarily involving cyclooxygenase (COX), lipoxygenase (LOX), and cytochrome P450 (CYP-450) enzymes [21].

COX is the key and rate-limiting enzyme responsible for the conversion of AA to PGs. There are two distinct isoforms of COX: COX-1, which is constitutively expressed in most tissues, and the inducible COX-2, which is expressed in cells of the TME, endothelial cells, smooth muscle cells, fibroblasts, chondrocytes, and cancer cells [61,62]. Several decades ago, researchers discovered a significant increase in COX-2 protein expression and activity in lymphoma cell lines [63]. In B-NHL, a correlation was identified between COX-2 upregulation and deletions on chromosome 17p, where the tumor suppressor gene TP53 is located [64], which is associated with poor outcomes [65]. COX-2 promotes apoptotic resistance, proliferation, transformation, angiogenesis, inflammation, invasion, and metastasis of malignant cells, mainly through its metabolite PGE2 [66,67,68]. Of interest is that platelets primarily express COX-1, which generates large amounts of thromboxane (TX)A2, a proaggregatory and vasoconstrictor mediator of chronic inflammation associated with several diseases, including tissue fibrosis and cancer [69].

Aspirin (acetylsalicylic acid), a non-selective COX inhibitor, has shown potential therapeutic effect as an adjunct to existing therapies in various cancers, including lymphoma [70,71]. Its anti-inflammatory properties, through the inhibition of COX-1 and COX-2, may reduce the inflammatory response within the TME, which contributes to tumorigenesis and immune evasion [72]. The role of aspirin in cancer prevention is still a matter of debate [73]. A large case-control study including 1703 NHL patients and 2199 frequency-matched controls, found that low-dose aspirin use was associated with a reduced risk of NHL, particularly DLBCL, with stronger protection observed with longer duration of use. This inverse association persisted, though slightly weakened, after adjusting for risk factors, including family history of NHL, body mass index, smoking, alcohol use, and rheumatoid arthritis, though it was slightly attenuated [74].

Selective COX-2 inhibitors like celecoxib have also been investigated in lymphoma therapy. These inhibitors selectively block COX-2 activity, without affecting COX-1, thus reducing unwanted side effects like gastrointestinal toxicity [75,76]. The role of celecoxib in improving chemotherapy efficacy in DLBCL has been explored, including in overcoming doxorubicin-induced multidrug resistance [77] and improving the effectiveness of CAR-T cell immunotherapy [78,79].

On the other hand, n-3 PUFAs, including eicosapentaenoic acid (EPA, 20:5) and docosahexaenoic acid (DHA, 22:6), have the opposite effect to those of n-6 PUFA. A higher ratio of n-6 to n-3 PUFAs in red blood cell membranes prior to diagnosis has been associated with an increased DLBCL risk [80]. The substantial decrease in n-3 PUFAs that corresponded with disease aggressiveness and poor prognosis suggested that n-3 PUFA levels may serve as a potential biomarker for NHL progression [39,40,81]. Supplementation with n-3 PUFAs has demonstrated beneficial effects in cancer patients by enhancing chemotherapy efficacy, mitigating treatment-related side effects, and thereby improving clinical outcomes and quality of life [82,83,84]. A meta-analysis of 20 studies involving 971 cancer patients receiving n-3 PUFA as adjuvant therapy showed a significant reduction in circulating proinflammatory cytokine levels [85]. The results of a recent meta-analysis of 28 clinical trials confirmed the positive anti-inflammatory effects of n-3 PUFA supplementation as an add-on treatment in cancer patients, highlighting that moderate doses (e.g., 850–900 mg/day) can achieve these benefits without causing any significant side effects [86]. In addition to modulating immune responses, n-3 PUFAs exert anti-tumor effects by inducing reactive oxygen species (ROS) production, which leads to oxidative stress and damage to cellular components [87]. Furthermore, n-3 PUFAs play a role in tumor suppression by inducing epigenetic modifications, such as DNA methylation, histone methylation, and acetylation of histones and p53. These changes influence apoptotic signaling pathways, including the RAS/ERK/C/EBPβ, PI3K/AKT/mTOR, ERK/AKT/NF-κB, and BCR/PI3K/NF-κB pathways [88,89,90,91].

## 3. Protein-Derived Inflammatory Parameters

Protein-derived inflammatory markers like C-reactive protein (CRP), albumin, fibrinogen, lactate dehydrogenase (LDH), beta-2 microglobulin (β2M), and interleukins (ILs) play an important role in assessing the inflammatory state of DLBCL, influencing prognosis, treatment decisions, and disease monitoring. Targeting these inflammatory markers and pathways offers potential therapeutic benefits. For instance, reducing inflammation through therapies targeting cytokines like IL-6 and tumor necrosis factor alpha (TNF-α) could improve clinical outcomes and enhance the effectiveness of chemotherapy and immunotherapy in DLBCL patients.

### 3.1. CRP

CRP is an acute-phase protein produced by the liver in response to inflammatory cytokines, particularly IL-6, which is secreted by macrophages and T cells. Due to its ease of testing through simple blood samples, CRP has become a widely used biomarker for assessing a patient’s inflammatory status. Its levels are often elevated in various malignancies, both solid tumors and hematological cancers, highlighting its potential as a prognostic marker for disease progression and treatment response [92]. In Hodgkin lymphoma, baseline levels of CRP have been shown to correlate directly with advanced stages of the disease and/or the presence of B symptoms. Elevated CRP levels are also considered predictors of treatment response, with persistent high CRP values during treatment indicating resistance to therapy [93].

In DLBCL, CRP was identified as an independent prognostic marker over a decade ago [94]. The patients with low CRP expression receiving R-CHOP chemotherapy exhibit significantly better therapeutic responses compared to those with high CRP levels, emphasizing the negative prognostic impact of elevated CRP [95,96]. The prognostic utility of CRP in DLBCL has been validated by two meta-analyses [97,98]. A recent meta-analysis that included data from 11 studies involving 2314 patients demonstrated that high baseline CRP levels were associated with lower OS in DLBCL patients [98]. Beyond its role as a stand-alone prognostic marker, CRP enhances the predictive accuracy of the R-IPI score in DLBCL patients [99,100].

Elevated CRP levels have been linked to an increased risk of venous thromboembolism (VTE), particularly deep vein thrombosis (DVT), in cancer patients [101]. It has been reported that CRP, along with other inflammatory parameters, can help identify DLBCL patients at risk for VTE, who may benefit from thromboprophylaxis [102].

Furthermore, CRP has shown value as a biomarker in CAR T-cell therapy. In a multicenter clinical study exploring the predictive factors of response to immunotherapy, elevated CRP levels at the time of infusion (e.g., ≥10 mg/L) were found to predict not only treatment response but also the occurrence of potential therapy-related toxicities [103]. Elevated CRP levels prior to CAR T-cell infusion have also been shown to predict the length of intensive care unit stay [104], further underscoring their relevance in both disease monitoring and treatment decision-making.

### 3.2. Fibrinogen

Fibrinogen is an acute-phase reactant that plays a pivotal role in clot formation and inflammation. Elevated fibrinogen levels have been linked to increased tumor cell proliferation, metastasis, and angiogenesis and have also been associated with poor prognosis in various malignancies [105]. In DLBCL, a significant correlation between fibrinogen levels and the presence of B symptoms and clinical stage has been reported [106]. Moreover, in a retrospective study, Niu et al. revealed that hyperfibrinogenemia was associated with higher NCCN-IPI scores, further highlighting its prognostic value [107].

Several studies have also demonstrated that elevated fibrinogen levels at presentation are linked to poor outcomes, including shorter PFS and OS in DLBCL patients [107,108,109], reinforcing its prognostic relevance. Additionally, Ogura et al. found that elevated fibrinogen levels prior to stem cell transplantation were associated with poorer survival outcomes, emphasizing the importance of this inflammatory marker in the management and prognosis of DLBCL [110].

### 3.3. Albumin

Albumin is a negative acute-phase protein whose levels decrease in response to systemic inflammation. Low serum albumin levels are associated with poor nutritional status, increased tumor burden, and a worse prognosis in cancer patients [111,112]. In DLBCL, hypoalbuminemia (defined as serum albumin < 35 g/L) has been identified as a significant predictor of poor prognosis in DLBCL [113,114]. Conversely, higher serum albumin levels (≥40 g/L) proved to be an independent prognostic factor of superior outcomes in DLBCL patients aged ≥ 70 years [115]. Hypoalbuminemia is usually considered a marker of malnutrition that is common in elderly population and cancer patients, which can worsen clinical outcome. To assess the nutritional status of patients, two indices that incorporate albumin levels, namely, the Prognostic nutrition index (PNI) and Geriatric nutritional index (GNRI), were originally introduced to identify at-risk medical patients. These indices have been extensively evaluated in patients with hematological malignancies and have proven to be valid predictors of prognosis [116,117,118,119]. In a cohort of 95 older DLBCL patients, low GNRI but not sarcopenia was significantly associated with adverse prognosis [120]. These findings underscore the value of albumin as a critical marker not only for nutritional assessment but also for predicting clinical outcomes, particularly in elderly populations who may experience increased inflammation and nutritional depletion due to both their age and cancer.

Albumin, along with CRP and fibrinogen, are routinely measured in most biochemical laboratories, making them affordable and accessible inflammatory biomarkers.

### 3.4. IL-6

Numerous cytokines are released during the inflammatory response, some exerting tumor-promoting effects while others play a tumor-suppressive role. A better understanding of the cytokine landscape has provided deeper insights into tumorigenesis. Members of the IL-6 cytokine family, including IL-6 itself, oncostatin M, leukemia inhibitory factor, IL-11, IL-27, IL-31, ciliary neurotrophic factor, cardiotrophin 1, and cardiotrophin-like cytokine factor 1, have been implicated in promoting tumor development by influencing the TME, as illustrated in Figure 2, underscoring their potential as therapeutic targets in oncology [121].

Among these cytokines, IL-6 stands out as a key mediator of the link between inflammation and cancer, and it is the most widely applicable and clinically evaluated cytokine for prognostication in routine clinical practice. IL-6 promotes tumor growth, immune evasion, and resistance to therapy by activating key oncogenic pathways, including JAK/STAT3 and NF-κB [122,123,124]. It also reshapes the tumor microenvironment by modulating immune cell populations and inducing the polarization of TAMs, which further enhances tumor growth and therapy resistance [125,126]. Elevated IL-6 levels are consistently associated with poor clinical outcomes in DLBCL. For instance, Bao et al. found that IL-6 levels ≥ 4.5 pg/mL were predictive of shorter PFS and OS [127]. Furthermore, high IL-6 levels were associated with resistance to treatment [128,129] and with clinical features including advanced disease stage and B symptoms [130]. These findings highlight the importance of IL-6 as a valuable marker in both risk stratification and monitoring therapeutic efficacy in DLBCL patients but also as possible therapeutic target [131]. An IL-6 receptor antagonist, tocilizumab, is used for prevention and treatment of cytokine-release syndrome, a potentially life-threatening complication during T-cell-directed treatment [132,133]. Furthermore, tocilizumab improves cytotoxic activity of antibody–drug conjugate naratuximab emtansine in CD37+ DLBCL cells [134].

### 3.5. β2M

β2M is an essential component of class I major histocompatibility complex and is present on all nucleated cells. Elevated levels of β2M in blood and bodily fluids are indicative of cell membrane turnover, cellular turnover rates, and kidney function and serve as a useful inflammatory marker in various cancers, particularly B-cell malignancies. The recent research, including a genome-wide meta-analysis (GWMA), bidirectional two-sample Mendelian randomization (TSMR), and pathway enrichment analysis, has established a causal link between increased β2M levels and a higher risk of DLBCL, potentially associated with innate immune system dysfunction [135]. Furthermore, a retrospective analysis involving 3232 newly diagnosed DLBCL patients treated with CIT, as recorded in the Danish Lymphoma Registry, demonstrated that incorporating β2M into the NCCN-IPI enhances the discriminatory performance of this prognostic tool [136].

## 4. Cells of Innate and Adaptive Immune Responses in the TME of DLBCL

Lymphomagenesis is characterized by complex interactions between tumor cells, innate immune cells, and adaptive immune cells within the TME. Both innate and adaptive immune responses are essential for controlling tumor growth, and the interactions between tumor cells and immune cells in the TME significantly impact clinical outcomes [137]. The polarization of macrophages from M1 to M2 [138]; the dysfunction of B cells [139], NK cells [140], and T-cells [141]; and the upregulation of ICP [142] all contribute to immune evasion and tumor progression, as displayed in Figure 3. Understanding these dynamics among immune cells offers insights into potential therapeutic strategies, such as ICPis, regulatory T-cell (Treg) depletion, and BsAb and CAR T-cell therapies, which could help restore immune surveillance and improve treatment outcomes in DLBCL [143].

### 4.1. Cells of the Innate Immune Response in DLBCL

#### 4.1.1. Monocytes and TAMs

TAMs, derived from monocytes circulating in the bloodstream, represent the most abundant immune cell populations within the TME and play a crucial role in the progression and prognosis of lymphoma [138]. Upon migration to tissues, monocytes differentiate into macrophages and polarize into two primary phenotypes depending on environmental signals from the TME: M1/classically activated and M2/alternatively activated. M1 macrophages, activated by pro-inflammatory signals like interferon-gamma (IFN-γ), IL-1β, and IL-12, exhibit anti-tumor activity by enhancing cytotoxic T-cell responses and directly killing tumor cells [144]. In contrast, M2 macrophages exert immunosuppressive functions, producing IL-10, transforming growth factor (TGF)-β, PD-L1, and vascular endothelial growth factor (VEGF), which support tumor progression, tissue remodeling, angiogenesis, and metastasis [145]. Additionally, M2 macrophages contribute to immune suppression by inhibiting T-cell function and enhancing tumor resistance to chemotherapy [146]. M2 macrophages also contribute to the degradation of the extracellular matrix through the overexpression of legumain, which further promotes the progression of DLBCL [147]. The polarization of macrophages from M1 to M2 is frequently observed in DLBCL and correlates with poor prognosis [148,149]. TAMs have distinct phenotypes and genetic signatures. M2 macrophages express CD163, while the expression of the pan-macrophage marker CD68 is upregulated in M1 compared to M2 macrophages [150]. A retrospective analysis of clinicopathologic data and survival outcomes from 355 patients diagnosed with DLBCL demonstrated that a CD163+ M2 TAM content of 9.5% or greater negatively affects the prognosis of DLBCL [151]. Guo et al. recently proposed the M2-macrophage-related gene prognostic model for DLBCL constructed on the expression of eight genes (MS4A4A, CCL13, LTB, CCL23, CCL18, XKR4, IL22RA2, and FOLR2). Elevated immune scores in high-risk DLBCL patients were associated with poorer prognosis and greater sensitivity to chemotherapeutic agents and ICPis [152]. Novel treatment strategies targeting TAMs to overcome tumor resistance are intensively being investigated. Both animal models and clinical trials focused on phagocytosis checkpoints, the reversal from M2 to M1 phenotype, and CAR-macrophage (CAR-M) have shown promising results with high efficacy and low toxicity [153].

Monocytes circulate in the peripheral blood for 1 to 3 days before migrating into tissues, where they differentiate into macrophages. The relationship between peripheral absolute monocyte count (AMC) and TAM in DLBCL remains a topic of ongoing research. Monocytosis at the time of DLBCL diagnosis has been identified as an independent prognostic factor associated with poorer outcomes and reduced OS. Wilcox et al., in a cohort of 366 DLBCL patients, demonstrated that an absolute monocyte count (AMC) above 0.63 × 10^9^/L at diagnosis is associated with poorer prognosis, with a 5-year OS rate of 59% compared to 71% in patients with lower AMC [154]. This cutoff was subsequently validated in a large multicenter study of 1017 therapy-naïve DLBCL patients from Israel and Italy, where it retained its negative prognostic value even after adjusting for the IPI (HR 1.54, *p* = 0.009) [155]. Additionally, patients with negative end-of- treatment (EOT)-PET-CT but AMC > 0.63 × 10^9^/L had significantly lower 3- and 5-year event-free survival (EFS) rates, suggesting that combining AMC with EOT-PET-CT strengthens the prediction of durability of treatment response [156].

#### 4.1.2. Neutrophils and Tumor Associated Neutrophils (TANs)

Neutrophils, the most abundant circulating leukocytes and primary responders to bacterial infections and tissue damage, also play a role in the tumor response in DLBCL. Within TME, pro-inflammatory cytokines such as IL-1, IFN-γ, and TNF-α stimulate the production of chemokines that recruit neutrophils [157]. These TANs promote tumor progression and immune suppression by releasing ROS and proteolytic enzymes that degrade the extracellular matrix, thereby facilitating tumor invasion. Neutrophils contribute to immune evasion by suppressing T-cell-mediated anti-tumor responses through the secretion of inhibitory cytokines like IL-10 and chemokines such as C-X-C motif chemokine ligand 8 (CXCL8), thereby facilitating tumor immune escape [158]. Another key mechanism is the formation of neutrophil extracellular traps (NETs)—web-like structures composed of decondensed chromatin—that can capture circulating tumor cells (CTCs) and enhance metastatic potential by promoting tumor cell survival [159,160].

In clinical settings, an elevated absolute neutrophil count (ANC) has been observed in patients with advanced-stage DLBCL (*p* = 0.012) and is significantly associated with poorer DFS [161]. While ANC alone has prognostic value, it is more commonly incorporated into composite indices alongside other hematologic and biochemical markers, which will be discussed later.

#### 4.1.3. NK Cells

NK cells are essential effectors of the innate immune system, responsible for eliminating tumor and virus-infected cells through distinct mechanisms, including direct cytotoxicity via the release of perforin and granzymes, as well as induction of apoptosis through death receptor pathways such as Fas ligand and TNF-related apoptosis-inducing ligand. They constitute approximately 10–15% of peripheral blood lymphocytes and are phenotypically defined by the expression of CD3^−^CD56^+^. NK cells are further categorized into two main subsets based on CD56 expression: CD56^bright^ NK cells, which are relatively immature, account for 5–10% of circulating NK cells and primarily exert immunoregulatory functions through cytokine secretion; and CD56^dim^ NK cells, which are fully mature, comprise 90–95% of peripheral NK cells and are mainly responsible for direct cytotoxicity, including Ab-dependent cell-mediated cytotoxicity (ADCC) [162,163].

In the TME, NK cells often become dysfunctional due to the upregulation of ICP receptors such as PD-1 and T-cell immunoglobulin and mucin-domain-containing protein 3 (TIM-3), which markedly reduce their cytotoxic potential [140,142,164]. Furthermore, macrophage polarization and the accumulation of Tregs contribute to NK cell suppression and immune evasion [165,166]. Clinically, elevated levels of circulating NK cells before treatment have been associated with improved therapeutic outcomes in relapsed/refractory (R/R) DLBCL [167]. Notably, NK cell count < 0.1 × 10^9^/L has been identified as an independent adverse prognostic factor for PFS and OS in the non-GCB subtype of DLBCL [168].

#### 4.1.4. DCs

Myeloid-derived DCs, which present tumor antigens and trigger naïve T-cell activation and differentiation, express the surface marker CD11c, linked to cytotoxic effects against lymphoma cells [169]. Decreased CD11c+ DC in the TME has been linked with MYC and BCL2/BCL6 (double-hit/triple-hit) genotype and survival in DLBCL [170,171]. In contrast, a higher density of DCs in TME [172] and an increased number of CD11c+ DCs in the peripheral blood of DLBCL patients have been linked with better prognosis and OS [173]. Lee et al. developed a prognostic model for patients with EN DLBCL receiving CIT, incorporating the expression of CD11c and Forkhead box P3 (FOXP3), a key transcription factor involved in the regulation of Treg cell function, within the TME [174]. However, the accurate assessment of peripheral NK cell and DCs levels typically requires flow cytometry, a method that is costly and impractical for routine clinical practice.

### 4.2. Cells of Adaptive Immune Response in DLBCL

Since DLBCL originates from B lymphocytes, dysregulation of B-cell receptor (BCR) signaling plays a pivotal role in its pathogenesis. Aberrant BCR signaling promotes tumor cell survival, enhances proliferation, and confers resistance to apoptosis, primarily through activation of the NF-κB and PI3K/AKT pathways [175]. The antitumor B-cell immune response is further compromised by immunosuppressive elements in the TME, particularly Tregs and TAMs. Tregs, which are essential for maintaining immune homeostasis, can suppress not only NK cell activity but also B-cell and CD4^+^/CD8^+^ T-cell responses when enriched in the TME of DLBCL [139].

T cells are central to adaptive immunity: CD8^+^ cytotoxic T cells directly eliminate target cells, while CD4^+^ helper T cells assist B cells in antibody production. T-cell activation is initiated when antigens presented on major histocompatibility complex (MHC) molecules are recognized by the T-cell receptor (TCR), leading to downstream signaling cascades. However, in approximately 50% of DLBCL cases, there is a loss of MHC class I expression on tumor cells, which may impair CD8^+^ T-cell recognition [176]. One of the primary antitumor functions of CD8^+^ T cells, and a key mechanism targeted by immunotherapies, is the induction of apoptosis via the perforin–granzyme pathway. CD4^+^ T cells also contribute significantly to antitumor immunity, and a TME enriched with CD4^+^ T cells (over 20%) has been associated with improved relapse-free survival and OS [177,178]. Similar to NK cells, both CD4^+^ and CD8^+^ T cells can become functionally exhausted within the TME. As noted earlier, tumor cells exploit ICP pathways, such as PD-1, PD-L1, CTLA-4, and FOXP3-expressing Tregs, to dampen T-cell activity and escape immune surveillance [141]. Recent studies have shown that elevated expression of OX40, a costimulatory receptor expressed on activated T cells, in DLBCL is associated with co-expression of ICP proteins and enhanced T-cell activation. This contributes to an inflamed TME, favorable clinical characteristics, and improved response to CIT, underscoring the potential of OX40 as both a prognostic biomarker [179] and a therapeutic target [180]. In a separate investigation of the mutational landscape of DLBCL, Zhang et al. identified 227 significantly mutated genes. Notably, tumor protein 53 (TP53) and CD58 mutations displayed a mutually exclusive pattern. The TP53WT and CD58MUT subgroup, especially within the GCB subtype, was associated with poor prognosis, increased immune evasion, and preferential expression of co-inhibitory receptors such as PD-1, TIM-3, and lymphocyte activation gene 3 (LAG-3). This profile suggests potential sensitivity to immunotherapy. Moreover, this subgroup exhibited elevated infiltration of exhausted T cells, macrophages, and NK cells, indicative of a distinctly immunosuppressive TME [181].

Patients with DLBCL exhibit reduced levels of peripheral blood CD4^+^, CD8^+^ T cells, and NK cells compared to healthy individuals. High-risk patients also show increased Treg proportions and diminished IFN-γ production by CD4^+^ and CD8^+^ T cells, indicating an impaired cellular immune response [182]. The prognostic significance of pretreatment lymphopenia (<1 × 10^9^/L) has been recognized across various cancers, including lymphoma, for decades [183]. A meta-analysis involving 1206 DLBCL patients from six studies further confirmed that low baseline absolute lymphocyte counts (ALC) are associated with poorer DFS and OS [184]. Lymphocyte count in peripheral blood serves as a key surrogate marker of immune reconstitution following autologous stem cell transplantation (ASCT) in NHL. Patients with an ALC on day 15 post-transplant (ALC-15) ≥ 0.5 × 10⁹/L demonstrated improved clinical outcomes [185]. Pre-lymphodepletion ALC has been identified as a predictor of response to CAR T-cell therapy. Patients with higher ALC prior to lymphodepletion showed significantly improved OS (45.17 vs. 9.6 months, *p* = 0.008) and PFS (45.17 vs. 4.07 months, *p* = 0.030) compared to those with lower ALC [186].

## 5. Role of Platelets and Anemia in Inflammation and DLBCL Prognostication

Beyond their role in hemostasis, platelets contribute to immune modulation by releasing cytokines and chemokines that attract neutrophils and monocytes and activate DCs, thereby influencing both innate and adaptive immunity [187]. In addition to immune cells, platelets activate fibroblasts and vascular cells through the release of various bioactive molecules, including lipid-derived eicosanoids, growth and angiogenic factors, and RNA-rich extracellular vesicles. During early tumorigenesis, platelet activation induces COX-2 overexpression in stromal cells, leading to increased PGE2 production, which, in turn, promotes cancer progression by inhibiting apoptosis, enhancing tumor cell proliferation and migration, and enabling immune evasion [188]. In DLBCL, pretreatment platelet count has emerged as a potential prognostic marker. In a study of 1007 patients treated with frontline R-CHOP or R-CHOP-like regimens, a platelet count ≤ 157 × 10^9^/L was independently associated with poorer OS (HR 1.960, 95% CI 1.418–2.709, *p* < 0.001) and PFS (HR 1.443, 95% CI 1.080–1.927, *p* = 0.013) [189].

Cancer-related anemia has a multifactorial origin, with its prevalence varying from 30% to 90% depending on the cancer type and site [190]. In follicular NHL, anemia is a well-established prognostic factor and is incorporated into standard risk models when hemoglobin (Hb) falls below 120 g/L [191,192]. However, in DLBCL, specific Hb thresholds with prognostic or predictive significance have yet to be clearly defined. According to Hong et al., a baseline Hb level below 100 g/L (determined as the lowest value within seven days before initiating R-CHOP, without prior red blood cell transfusion) was significantly associated with shorter DFS and OS in a cohort of 157 DLBCL patients [193]. Furthermore, Matsumoto et al., proposed a simple prognostic index based on grade 2 anemia (defined as Hb < 100–80 g/L) and advanced Ann Arbor clinical stages (CS) to predict outcome in DLBCL patients [194].

## 6. Inflammatory Indices Evaluated in DLBCL

Although developed in the pre-rituximab era, the IPI remains the most widely used tool for risk stratification in newly diagnosed DLBCL patients across all age groups. Other proposed prognostic indices, such as R-IPI, age-adjusted IPI (aa-IPI), and NCCN-IPI are all derived from the same clinical variables as the original IPI and none can accurately identify patients at ultra-high risk [195]. Efforts to enhance prognostic precision, such as the IPI-IPM, which incorporates molecular markers including CMBL, TLCD3B, SYNDIG1, ESM1, EPHA3, HUNK, PTX3, and IL12A, have shown promise [196], but their complexity limits their routine clinical application.

### 6.1. Prognostic Indices That Include Inflammatory Biomarkers

The success of immune-based therapies, coupled with growing insights into the impact of systemic inflammation and the TME on DLBCL progression, treatment response, and survival, has driven efforts to enhance the traditional IPI. Due to their established prognostic relevance, routinely available lipid-derived (e.g., HDL-C, apoA) and protein-based (e.g., CRP, albumin, hemoglobin, fibrinogen, and β2M) inflammatory markers have been used either as standalone biomarkers, as previously discussed, or more commonly integrated into existing prognostic models—or combined to create novel indices that are easily applicable in everyday clinical practice.

Most of the developed indices include albumin as a key component. Both the pretreatment albumin-to-globulin ratio and the CRP-to-albumin ratio have been associated with poorer prognosis in patients with DLBCL [197,198]. Additionally, the CRP-to-albumin ratio and albumin-to-fibrinogen have been suggested as a potential complement to the IPI for improved risk stratification in DLBCL [199,200]. In addition, more complex indices have also been developed. For instance, Melchardt et al. refined the NCCN-IPI for elderly DLBCL patients by incorporating low serum albumin and β2M, enhancing its prognostic accuracy [201]. Similarly, Gang et al. integrated hypoalbuminemia with traditional IPI variables to develop age-adjusted prognostic indices with a 70-year cutoff, improving risk stratification [202]. The Kyoto Prognostic Index (KPI), which combines LDH, albumin, ECOG-PS, and EN involvement, has demonstrated superior predictive power in the rituximab era [203]. Additionally, GNRI, calculated from serum albumin and body weight ratios, has been identified as an independent prognostic marker reflecting patients’ nutritional status [204].

The key components and principal prognostic and predictive findings of several proposed models containing lipid- and protein-derived markers are summarized in Table 1.

### 6.2. Prognostic Indices That Include Peripheral Blood Cells

Peripheral blood cells (PBCs)—including lymphocytes, monocytes, neutrophils, and platelets—not only reflect systemic immune and inflammatory responses but also mirror the biological activity of the TME, as we have described earlier in this review. Their prognostic significance in DLBCL is well established, and they have become central components of several novel prognostic models that we will further analyze. These indices, whether based on ratios of absolute PBC counts or integrated with other inflammatory markers, substantially improve risk stratification and offer a more accurate and comprehensive prediction of clinical outcomes in newly diagnosed DLBCL.

Among these, platelet count and ALC are most frequently utilized in the composite prognostic indices. Some indices are based on two variables, such as LAR (LDH to ALC ratio) [200], PNI (albumin and ALC) [117], PA (platelet and albumin) [205], and HP (Hb and platelet) [206]. Others, like the model proposed by Chen H. et al., incorporate three components—β2M, platelet count, and red cell distribution width (RDW) [207]. In an effort to better identify high-risk patients, some indices integrate multiple blood cell types. For example, Chen et al. developed a composite model incorporating baseline ANC, AMC, β2M, Eastern Cooperative Oncology Group Performance Scale (ECOG PS), and number of extranodal (EN) sites. This model demonstrated superior risk discrimination compared to the standard IPI [208]. Jelicic et al. recently proposed the Elderly-IPI (E-IPI), which includes all NCCN-IPI parameters (excluding EN involvement), along with serum albumin and platelet counts to enhance prognostic precision in elderly patients [209].

Indices based on ratio of PBCs, such as lymphocyte-to-monocyte ratio (LMR), neutrophil-to-lymphocyte ratio (NLR), and platelet-to-lymphocyte ratio (PLR) have demonstrated strong predictive value in both solid tumors and hematological malignancies [210,211]. These ratios are also proven to be effective in predicting treatment response and survival outcomes in DLBCL [212,213,214]. More complex models combining three or more blood cell types have also been developed [215,216], while others integrate blood cell ratios with additional inflammatory markers [213], as detailed in Table 2.

Some of these indices have been validated through systematic reviews and meta-analyses. A meta-analysis of studies involving 1311 patients demonstrated that a low Prognostic Nutritional Index (PNI) is an independent indicator of poor prognosis in DLBCL, associated with dismal prognosis and poor DFS and OS [217]. Further supporting the clinical relevance of nutritional and inflammatory markers, two separate systematic reviews assessed the GNRI. One analysis of 2448 elderly patients with DLBCL showed that lower GNRI values were significantly linked to reduced DFS and OS [218]. Another review, based on data from 2353 patients, confirmed that low GNRI not only predicts inferior survival outcomes but is also correlated with poor PS, advanced stage, and the presence of B symptoms [219].

Regarding indices incorporating PBC ratios, a meta-analysis of 1931 DLBCL patients found that elevated PLR was significantly associated with inferior OS and adverse clinical features such as B symptoms, elevated LDH, advanced stage, and ECOG PS ≥ 2, though it showed no association with PFS, age, gender, or cell of origin [220]. Likewise, a systematic review of 11 studies involving 4578 patients consistently identified low LMR as a strong predictor of reduced OS and DFS, reinforcing its prognostic relevance in DLBCL [221].

Beyond meta-analyses, direct comparisons of various prognostic indices within the same cohort have also been conducted. For example, Wang et al. demonstrated that the systemic immune–inflammation index (SII) outperformed both PLR and NLR in predicting outcomes in 224 newly diagnosed DLBCL patients [216]. In R/R DLBCL, Kim et al. evaluated multiple inflammation-based models: NLR, derived NLR (dNLR), LMR, PLR, Glasgow Prognostic Score (GPS), PNI, systemic inflammation response index (SIRI), and SII, alongside NCCN-IPI. Their findings identified high NCCN-IPI, GPS 2, and elevated dNLR as independent predictors of poor prognosis, leading to the development of a new risk model for PFS and OS [222].

Most recently, Ide D. et al. introduced the Kyoto Prognostic Index for relapsed/refractory DLBCL, which integrates low LMR, elevated LDH, and high CRP. This model showed promising predictive power for treatment response and survival in transplant-ineligible patients [223].

## 7. Conclusions and Further Directions

This narrative review highlights the critical role of inflammation in the pathogenesis and prognosis of DLBCL, with a particular emphasis on its incorporation into prognostic indices. A range of inflammation-based markers has been proposed as accessible, cost-effective tools designed to bridge the gap between conventional clinical scoring systems and complex, resource-intensive molecular models, thereby enhancing the accuracy of DLBCL risk stratification. While many of these models appear promising, most have been derived from retrospective, single-center cohorts with limited sample sizes and lack rigorous external validation. To incorporate inflammatory markers into routine clinical practice, several challenges must be addressed, including standardization of measurement methods, establishment of uniform cut-off values, and controlling for potential confounding factors.

Notably, a comparative analysis of 13 inflammation-based prognostic models in over 5000 patients from the Danish lymphoma registry found no advantage over the established NCCN IPI [224]. This highlights the gap between theoretical promise and clinical applicability. To advance personalized care in DLBCL, these inflammation-driven indices require rigorous validation through large-scale, prospective, multicenter prospective studies, and randomized clinical trials. Only with such high-quality evidence can these markers be confidently integrated into routine prognostic assessments and therapeutic decision making, firmly establishing their role as reliable and useful tools in clinical practice.

## Figures and Tables

**Figure 1 ijms-26-05670-f001:**
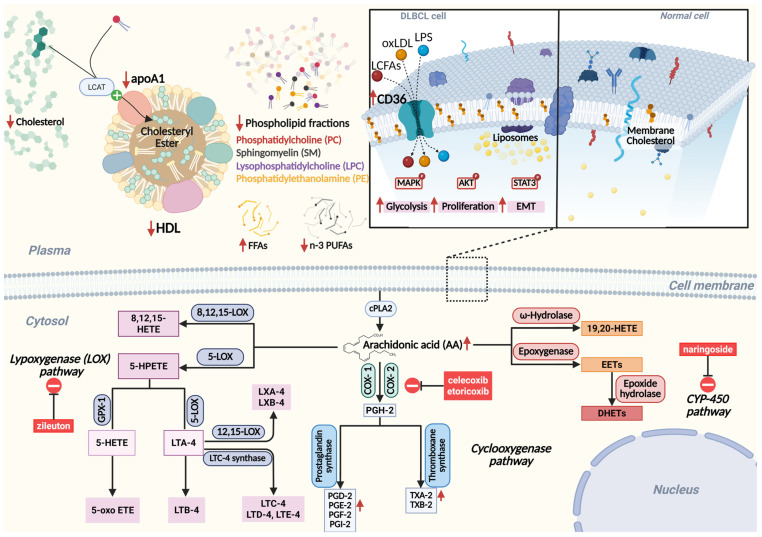
Altered lipid metabolism and arachidonic acid (AA)-derived signaling pathways in diffuse large B-cell lymphoma (DLBCL). DLBCL is associated with decreased HDL and apoA1, impaired cholesterol efflux, and altered phospholipid composition, including reductions in PC, SM, LPC, and PE. This leads to accumulation of FFAs and decreased n-3 PUFAs. CD36-mediated uptake of oxLDL, LPS, and LCFAs activates MAPK, AKT, and STAT3 signaling, promoting glycolysis, proliferation, and EMT. Elevated intracellular AA, released by cPLA2, is metabolized via three key enzymatic routes: the COX pathway (producing PGs and TXs), the LOX pathway (yielding HETEs, LTs, and lipoxins), and the CYP450 pathway (producing EETs and HETEs). These pathways collectively contribute to inflammation, angiogenesis, and tumor progression. Explanation: ↑, increased; ↓, decreased. Abbreviations: HDL, high-density lipoprotein; LCAT, lecithin–cholesterol acyltransferase; apoA1, apolipoprotein A1; FFAs, free fatty acids; PUFAs, polyunsaturated fatty acids; oxLDL, oxidized low-density lipoprotein; LCFAs, long-chain fatty acids; LPS, lipopolysaccharide; CD36, cluster of differentiation 36; MAPK, mitogen-activated protein kinase; AKT, protein kinase B; STAT3, signal transducer and activator of transcription 3; EMT, epithelial–mesenchymal transition; LOX, lipoxygenase; 5-LOX, 5-lipoxygenase; 12,15-LOX, 12/15-lipoxygenase; 8,12,15-LOX, 8/12/15-lipoxygenase; 5-HPETE, 5-hydroperoxyeicosatetraenoic acid; 5-HETE, 5-hydroxyeicosatetraenoic acid; 8,12,15-HETE, 8,12,15-hydroxyeicosatetraenoic acid; 5-oxo ETE; 5-oxo-eicosatetraenoic acid; LTA-4, leukotriene A4; LTB-4, leukotriene B4; LTC-4, leukotriene C4; LTD-4, leukotriene D4; LTE-4, leukotriene E4; LXA-4, lipoxin A4; LXB-4, lipoxin B4; GPX-1, glutathione peroxidase 1; COX-1, cyclooxygenase 1; COX-2, cyclooxygenase 2; cPLA2, cytosolic phospholipase A2; PGH-2, prostaglandin H2; PGD-2, prostaglandin D2; PGE-2, prostaglandin E2; PGF-2, prostaglandin F2; PGI-2, prostaglandin I2; TXA-2, thromboxane A2; TXB-2, thromboxane B2; CYP-450, cytochrome P450; EETs, epoxyeicosatrienoic acids; DHETs, dihydroxyeicosatrienoic acids; 19,20-HETE, 19,20-hydroxyeicosatetraenoic acid.

**Figure 2 ijms-26-05670-f002:**
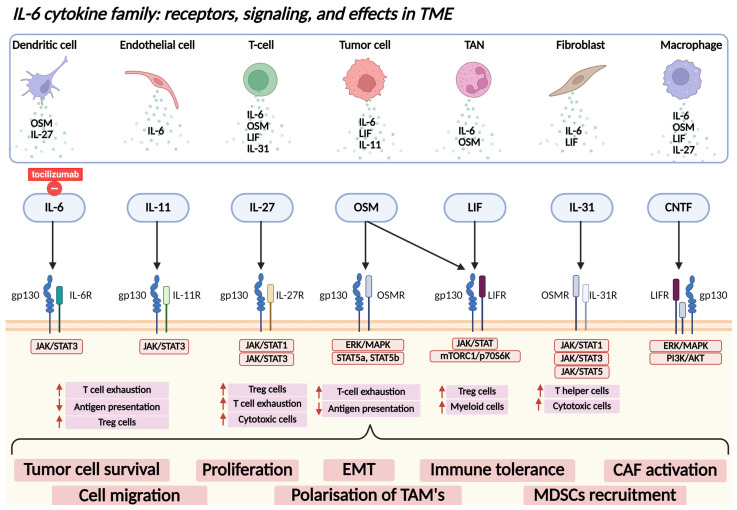
IL-6 cytokine family signaling in the tumor microenvironment (TME). Various components of the TME, including dendritic cells, T cells, macrophages, endothelial cells, tumor-associated neutrophils (TANs), fibroblasts, and tumor cells, secrete members of the IL-6 cytokine family, such as IL-6, IL-11, IL-27, oncostatin M (OSM), leukemia inhibitory factor (LIF), IL-31, and ciliary neurotrophic factor (CNTF). These cytokines signal through specific receptor complexes that typically include glycoprotein 130 (gp130) in combination with ligand-specific receptor subunits. Ligand binding triggers the activation of canonical intracellular signaling pathways, which collectively mediate immunosuppressive functions within the TME and contribute to tumor cell survival, proliferation, and immune evasion. Explanation: ↑, increased; ↓, decreased. Abbreviations: IL-6, interleukin-6; IL-11, interleukin-11; IL-27, interleukin-27; IL-31, interleukin-31; CNTF, ciliary neurotrophic factor; OSM, oncostatin M; LIF, leukemia inhibitory factor; IL-6R, interleukin-6 receptor; IL-11R, interleukin-11 receptor; IL-27R, interleukin-27 receptor; IL-31R, interleukin-31 receptor; LIFR, leukemia inhibitory factor receptor; OSMR, oncostatin M receptor; gp130, glycoprotein 130; JAK, Janus kinase; STAT, signal transducer and activator of transcription; ERK, extracellular signal-regulated kinase; MAPK, mitogen-activated protein kinase; PI3K, phosphoinositide 3-kinase; AKT, protein kinase B; mTORC1, mammalian target of rapamycin complex 1; p70S6K, p70 ribosomal protein S6 kinase; Treg, regulatory T cell; Th, T helper cell; Tc, cytotoxic T cell; TAM, tumor-associated macrophage; TAN, tumor-associated neutrophil; MDSC, myeloid-derived suppressor cell; CAF, cancer-associated fibroblast; TME, tumor microenvironment; EMT, epithelial-to-mesenchymal transition.

**Figure 3 ijms-26-05670-f003:**
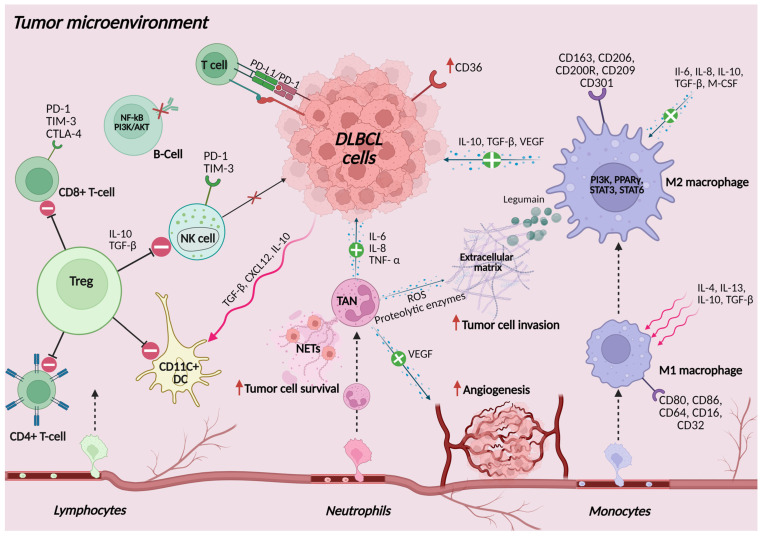
Tumor microenvironment (TME) in diffuse large B-cell lymphoma (DLBCL). The tumor microenvironment in DLBCL is characterized by immunosuppressive signaling, enhanced angiogenesis, and impaired anti-tumor immunity. DLBCL cells express programmed cell death protein PD 1 promoting T cell exhaustion through PD-1/PD-L1 interaction. Regulatory T cells and M2 macrophages secrete IL-10, TGF-β, and VEGF, which inhibit CD4^+^, CD8^+^, and NK cells and stimulate neovascularization. TANs promote tumor invasion by releasing ROS, proteolytic enzymes, and NETs. M2 macrophages further support tumor progression, with their polarization induced by IL-4 and IL-13. Dendritic cell function is suppressed, leading to reduced T cell activation. Additionally, BCR signaling in B cells is dysregulated through activation of the NF-κB and phosphoinositide 3-PI3K/AKT pathways, contributing to immune evasion and lymphoma cell survival. Explanation: ↑, increased; ↓, decreased. Abbreviations: Abbreviations: TME, tumor microenvironment; CD36, cluster of differentiation 36; CD4, cluster of differentiation 4; CD8, cluster of differentiation 8; CD16, cluster of differentiation 16; CD32, cluster of differentiation 32; CD64, cluster of differentiation 64; CD80, cluster of differentiation 80; CD86, cluster of differentiation 86; CD163, cluster of differentiation 163; CD200R, cluster of differentiation 200 receptor; CD206, cluster of differentiation 206; CD209, cluster of differentiation 209; CD301, cluster of differentiation 301; PD-1, programmed cell death protein 1; PD-L1, programmed death-ligand 1; TIM-3, T-cell immunoglobulin and mucin-domain containing molecule 3; CTLA-4, cytotoxic T lymphocyte-associated protein 4; Treg, regulatory T cell; CD11c⁺DC, CD11c-positive dendritic cell; NK cell, natural killer cell; NETs, neutrophil extracellular traps; TAN, tumor-associated neutrophil; ROS, reactive oxygen species; VEGF, vascular endothelial growth factor; TGF-β, transforming growth factor beta; IL, interleukin; TNF-α, tumor necrosis factor alpha; CXCL12, C-X-C motif chemokine ligand 12; PI3K/AKT, Phosphoinositide 3-kinase/protein kinase B; NF-κB, Nuclear Factor kappa B; STAT3, signal transducer and activator of transcription 3; STAT6, signal transducer and activator of transcription 6; PPARγ, Peroxisome Proliferator-Activated Receptor Gamma; M-CSF, macrophage colony-stimulating factor.

**Table 1 ijms-26-05670-t001:** Prognostic indices in DLBCL incorporating lipid- and protein-derived inflammatory markers.

Index Author, Year	Parameters Included	*N*	Cut Off/Prognostic Groups	Key Findings
Matsumoto Model Matsumoto et al., 2018 [194]	CS ≥ III Hb < 100 g/L	185	Scores 0, 1, 2	3-year PFS—score 0: 89.1%, score 1: 73.9%, score 2: 35.5%, (*p* < 0.001) 3-year OS—score 0: 94.6%, score 1: 82.0%, score 2: 61.4%, (*p* < 0.001)
Lipo-PI Gao et al., 2018 [30]	Concurrently: HDL-C <1.03 mmol/L LDL-C <2.60 mmol/L and NCCN IPI variables *	550 (T: 367, V: 183)	L: score 0–2 LI: score 3–4 HI: score 5–6 H: ≥7	5-year PFS—L: 88.6%, LI: 65%, HI: 29.5%, H: 10% 5-year OS—L: 96.9%, LI: 79.6%, HI: 45%, H: 22.5% Lipo-PI widened the definition of high-risk patients for 5-year OS and improved. the risk stratification of NCCN-IPI. HDL-C or LDL-C elevations after treatment correlated with better survival
IPI-A Yu et al., 2023 [28]	ApoA-I < 0.81 g/L and IPI variables **:	105	Comparison of IPI-A with IPI using a ROC curve	PFS (1 year: AUC = 0.745, 3 years: AUC = 0.827, 5 years: AUC = 0.763) A significant increase in TG, LDL-C, HDL-C, ApoA-I, and ApoB levels after Chemotherapy
AGR Yue et al., 2018 [197]	Albumin-to-globulin	335	1.3	Low AGR is an independent adverse predictor for OS (HR = 0.613; 95% CI = 0.412–0.910, *p* = 0.015) AGR distinguished patients with different prognosis in stage III–IV and the ABC DLBCL groups
GPS (Glasgow Prognostic score) Li et al., 2015 [198]	CRP > 10 mg/L Albumin < 35 g/L	160	GPS-0 GPS-1 GPS-2	Lower GPS predict better outcome, including PFS (*p* < 0.001) and OS (*p* < 0.001)
CAR Jung et al., 2021 [199]	CRP-to-albumin ratio	186	0.158	Low CAR is an independent adverse predictor for CRR (64.4% vs. 92.6%), 3-year PFS (53.5% vs. 88.0%). 3-year OS (68.3% vs. 96.2%)
AFR Shi et al., 2024 [200]	Albumin-to-fibrinogen	74	12.64	2-year PFS: high AFR-85.7%, low AFR-22.2% AFR can be used as good supplements for IPI to predict the prognosis of DLBCL
Modified NCCN-IPI Melchardt et al., 2015 [201]	Albumin < 35 g/L β2M > 3.0 mg/L NCCN-IPI variables *	499	L: score 0–2 LI: score 3 HI: score 4–7 H: score 8–10	3-year OS—L: 97.8%, LI: 82.7%, HI: 65.9%, H: 4.2% 5-year OS—L:93%, LI:78%, HI: 55.7%, H: 36.8%
DLBCL-PI (Modified DLBCL prognostic index) aaDLBCL-PI (for age ≤70 years) Gang et al., 2015 [202]	Albumin < 35 g/L and IPI variables ** (except EN) Albumin <35 g/L and IPI variables ** (except CS)	1990	L: score0–1 LI: score 2 HI: score 3 H: score 4–5	5-year OS in four risk groups—IPI: 83%, 64%, 57%, and 38% DLBCL-PI: 87%, 69%, 53%, and 37% aaDLBCL-PI: 92%, 84%, 74%, and 47%.
KPI (Kyoto Prognostic Index) Kobayashi et al., 2016 [203]	LDH ratio >1–3/>3, ECOG-PS ≥ 2, Albumin < 35 g/L EN (BM, bone, skin and/or lung/pleura)	323	L: score 0, LI: score 1–2, HI: score 3, H: score 4–5	3-year OS—L: 96.4%, LI: 84.7%, HI: 63.8%, H: 33.3% 3-year PFS—L: 84.4%, LI: 70.2%, HI: 53.4%, H: 24.1%
GNRI (Geriatric nutritional index) Atas et al., 2023 [204]	Lorentz formula ***	206	L and H groups according to the ROC curve (cut off 104.238)	Low GNRI is an independent adverse predictor DFS and OS

Abbreviations: HDL-C, high density cholesterol; LDL-C, low density cholesterol, TG, triglycerides, Apo, apolipoprotein; CRP, C-reactive protein; LDH, lactate dehydrogenase, EN, extranodal sites; IPI, International Prognostic Score; CS, Ann Arbor Clinical Stage; ECOG-PS, Eastern Cooperative Oncology Group Performance Scale; T, training cohort; V, validation cohort; L, low risk; LI, low–intermediate risk; HI, high–intermediate risk; H, high risk; ROC, receiver operating characteristic; AUC, area under the curve; CRR, complete remission rate; PFS, progression free survival, DFS, disease free survival; OS, overall survival. * NCCN-IPI variables: Age 1: ≤40–1, 2: 41–60, 3: 61–75; 4: >75, CS III/IV, ECOG PS > 1, EN, LDH 1: 1–3UNL, 2: >3UNL; ** IPI variables: Age > 60 years, CS III/IV, ECOG PS > 1, EN > 1, LDH > ULN; *** Lorenz formula [1.489 × serum albumin (g/L)] + [41.7 × (current body weight/ideal body weight)].

**Table 2 ijms-26-05670-t002:** Prognostic indices in newly diagnosed DLBCL incorporating peripheral blood cells.

Index Author, Year	Parameters Included	*N*	Cut Off/Prognostic Groups	Key Findings
LAR Shi et al., 2024 [200]	LDH to ALC ratio	74	244.95	2-year PFS—low LAR: 78.6%, high LAR:13.8% LAR can be used as good supplements for IPI to predict the prognosis of DLBCL.
PNI (Prognostic nutritional index) Zhou et al., 2016) [117]	albumin (g/L) + 5 × ALC × 10^9^/L	253	L and H groups according to the ROC curve	Low PNI was an independent adverse predictor EFS and OS
PA (Platelet–albumin) Ochi et al., 2017 [205]	platelet count <100 × 10^9^/L, albumin ≥ 35 g/L	391	L/I/H	5-year OS—L: 77.6%, I: 47.9%, H: 19.0%
HP (Hb–platelet) Nakayama et al., 2019 [206]	Hb < 120 g/L platelet counts <135 × 10^9^/L	89	L: score 0 I: score 1 H: score 2	Higher HP index was associated with worse OS and predicted OS rate independently of the IPI
Chen et al., 2022 [207]	β2M UNL platelets <157 × 10^9^/L RDW ≥ 14.5%	998 (T:701, V: 297)	L: score 0 LI: score 1 HI: score 2–3 H: score 4 VH: score ≥ 5	The AUC of the new model for OS prediction at specific time points (6 months to 10 years) was consistently higher than that of conventional prognostic models in both T and V
Chen et al., 2016 [208]	ANC AMC ECOG PS β2M EN	817	L1: score 0 L2: score 1 I: score 2 H1: score 3–5 H2: score 6–7	Better stratifies patients into various risk categories than the IPI for newly diagnosed DLBCL.
E-IPI (Elderly-IPI) Jelicic et al., 2025 [209]	NCCN-IPI variables * (excluding EN), albumin platelet	2835	L/LI/HI/H groups	3-year OS—L: 88.6%, H: 45% 5-year OS—L: 80.6%, H: 38.3% Δ c-index OS: 3–0.024 (0.021; 0.026)
LMR Markovic et al., 2014. [212]	Lymphocyte-to-monocyte ratio ≤ 2.8	222	L/H	L group had significantly shorter EFS and OS
PLR Zhao et al., 2018 [213]	Platelet-to-lymphocyte ratio ≥ 170	309	L/H	PLR was a significant prognostic factor for OS (*p* < 0.001) and PFS (*p* = 0.003)
NLR Wang et al., 2016 [214]	Neutrophil-to-lymphocyte ratio	156	3.0 L and H groups according to the ROC curve	5-year OS—L: −82.5%, H: 57.5% 5-year PFS—L: 64.5%, H: 30.0%
SIRI-PI (Systemic inflammation response index) Chu et al., 2023 [215]	Neutrophil × monocyte/ lymphocyte	153 PGl-DLBCL (T: 102, V: 51)	1.34 (L/H)	more precise high-risk assessment compared to NCCN-IPI with a higher AUC (0.916 vs. 0.835) and C-index (0.912 vs. 0.836) for OS
SII (Systemic immune–inflammation index) Wang et al., 2021 [216]	Neutrophil, Platelet, and lymphocyte counts	224	1046.1 L and H groups according to the ROC curve	3-year OS—L: 88.3%, H: 58.6% 3-year PFS—L: 74.4%, H: 3.7% SII superior to NLR and PLR
Zhao et al., 2018 [213]	PLR < 170 vs. ≥170 IPI < 2 vs. ≥2 or aaIPI 0 vs. ≥1 β2M normal vs. UNL	309	L/I/H	5-year OS—L: 86.4%, I: 54.1% H: 21.1%

Abbreviations: LDH, lactate dehydrogenase, ALC, absolute lymphocyte count; Hb, hemoglobin β2M, beta-2 microglobulin; RDW, red blood cell distribution width; ANC; absolute neutrophil count; AMC, absolute monocyte count; ECOG PS, Eastern Cooperative Oncology Group Performance Scale; EN, extranodal sites; PLR, platelet-to-lymphocyte ratio; IPI, International prognostic index, PGl, primary gastrointestinal: UNL, upper normal limit, T, training cohort; V, validation cohort; L, low risk; LI, low–intermediate risk; HI, high–intermediate risk; H, high risk, VH, very high risk; OS, overall survival; EFS, event free survival; * NCCN-IPI variables: Age 1: ≤40–1, 2: 41–60, 3: 61–75; 4: >75, CS III/IV, ECOG PS > 1, EN, LDH 1: 1–3UNL, 2: >3UNL.

## Data Availability

The data that are discussed in this article are presented in cited studies.

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
