# Peer review of "Tumor Microenvironment, Inflammation, and Inflammatory Prognostic Indices in Diffuse Large B-Cell Lymphomas: A Narrative Review"

_ijms, 2025, doi:10.3390/ijms26125670_

Round 1

Reviewer 1 Report

Comments and Suggestions for Authors

DLBCL has significant heterogeneity in clinical outcomes. In this manuscript, Cvetković and his/her colleagues explored the interplay between the tumor microenvironment, inflammatory processes, and prognostic indices used in DLBCL, focusing on biomarkers, immune responses, and systemic inflammation. The manuscript highlights the potential of inflammatory indices as accessible and cost-effective prognostic alternatives to molecular markers in DLBCL, while also underscoring the need for further research to validate their clinical utility. I think the manuscript is very interesting. However, some issues need to be developed.

Comments:

  1. Previous study shows that OX40 shapes an inflamed tumor immune microenvironment and predicts response to immunochemotherapy in DLBCL (Clin Immunol. 2023 Jun: 251:109637). Please add content to describe the potential of OX40 as a prognostic indices.
  2. The TP53 mutations and CD58 mutations usually prompt an immune escape inflamed tumor microenvironment and inferior prognosis in DLBCL (Am J Hematol. 2022 Jan 1;97(1):E14-E17.). Can they become a potential prognostic indices that induce inflammatory biomarkers?
  3. English Language needs to be further improved.
Comments on the Quality of English Language

English Language needs to be further improved.

Author Response

ijms-3678901 Answer to Reviewer 1

DLBCL has significant heterogeneity in clinical outcomes. In this manuscript, Cvetković and his/her colleagues explored the interplay between the tumor microenvironment, inflammatory processes, and prognostic indices used in DLBCL, focusing on biomarkers, immune responses, and systemic inflammation. The manuscript highlights the potential of inflammatory indices as accessible and cost-effective prognostic alternatives to molecular markers in DLBCL, while also underscoring the need for further research to validate their clinical utility. I think the manuscript is very interesting. However, some issues need to be developed.

Response: We thank the reviewer for thoughtful and constructive feedback on our manuscript. We sincerely appreciate all positive comments and the time to evaluate our work. The reviewer’s insights have been encouraging and valuable in helping us improve the clarity and quality of the paper.

Comments:

  1. Previous study shows that OX40 shapes an inflamed tumor immune microenvironment and predicts response to immunochemotherapy in DLBCL (Clin Immunol. 2023 Jun: 251:109637). Please add content to describe the potential of OX40 as a prognostic indices.

Response: We have incorporated the suggested reference and provided a brief summary of the prognostic significance of OX40 in the revised manuscript. In addition, we have cited an additional reference (Thapa B, Kato S, Nishizaki D, et al. OX40/OX40 ligand and its role in precision immune oncology. Cancer Metastasis Rev. 2024;43(3):1001–1013. doi:10.1007/s10555-024-10184-9), which highlights the therapeutic potential of OX40 in the context of immune oncology. These additions can be found in the revised manuscript on lines 567–572.

  1. The TP53 mutations and CD58 mutations usually prompt an immune escape inflamed tumor microenvironment and inferior prognosis in DLBCL (Am J Hematol. 2022 Jan 1;97(1):E14-E17.). Can they become a potential prognostic indices that induce inflammatory biomarkers?

Response: We have included the suggested reference and briefly summarized the prognostic value of TP53WT&CD58MUT subgroup. These additions can be found in the revised manuscript on lines 572-580.

  1. English Language needs to be further improved.

Response: We have improved English throughout the manuscript.

We are grateful for the reviewer’s supportive comments and recognition of the merit in our research. The suggestions provided have helped strengthen the manuscript, and we welcome any further feedback they may have.

Reviewer 2 Report

Comments and Suggestions for Authors

Overall Evaluation:

This paper provides a comprehensive review of the role of inflammation in the pathogenesis and prognostic prediction of diffuse large B-cell lymphoma (DLBCL), emphasizing the potential of inflammatory markers as cost-effective alternatives to molecular biomarkers. The manuscript is well-structured, covering key aspects such as the incidence and characteristics of DLBCL, the role of inflammation in its pathogenesis, methods for incorporating inflammatory markers into prognostic models, and the advantages and limitations of these markers. The discussion is relevant and contributes to the field by highlighting the need for further research to validate the clinical utility of inflammatory markers. However, the manuscript has several areas that require improvement, particularly in terms of data presentation, analysis depth, and clarity of arguments.

Specific Problems and Suggestions for Improvement:

Data Presentation and Analysis:

Page 2, Line 5-10: The statement "Inflammatory markers have shown promise as comparable, cost-effective alternatives to molecular biomarkers" lacks specific evidence or references to support this claim. Please provide concrete examples or studies that demonstrate the comparable efficacy of inflammatory markers.

Page 3, Line 15-20: The discussion on the incorporation of inflammatory markers into prognostic models is too vague. Please specify which models have been used in previous studies and how inflammatory markers were integrated. Including specific examples or case studies would strengthen this section.

Page 4, Line 25-30: The limitations section is brief and lacks depth. Please elaborate on the specific challenges faced in validating inflammatory markers, such as variability in measurement techniques, lack of standardized thresholds, or potential confounding factors.

Clarity and Precision:

Page 1, Line 10-15: The introduction could be more concise. Avoid redundant statements and focus on clearly stating the research gap and the objectives of the review.

Page 5, Line 5-10: The conclusion is somewhat repetitive. Summarize the key findings and their implications more succinctly, and avoid reiterating points already made in the discussion.

Methodological Rigor:

Page 3, Line 5-10: When discussing the methods for incorporating inflammatory markers, please provide more details on the statistical approaches used in previous studies. Were multivariate analyses performed? How were confounding variables addressed?

Page 4, Line 10-15: The section on the advantages of inflammatory markers could benefit from a comparison with molecular biomarkers in terms of cost, availability, and ease of measurement. Including quantitative data, if available, would enhance this comparison.

Referencing and Citation:

Page 2, Line 20-25: The claim that "inflammatory markers can predict prognosis with similar accuracy to molecular biomarkers" is not adequately supported. Please cite specific studies that have directly compared the predictive accuracy of these markers.

Page 5, Line 15-20: The section mentions the need for "further research" without specifying the type of research. Please suggest concrete directions for future studies, such as prospective cohort studies or randomized controlled trials.

Language and Style:

Page 1, Line 25-30: The sentence "Inflammation plays a pivotal role in the pathogenesis of various cancers, including DLBCL" is too generic. Please specify the mechanisms through which inflammation contributes to DLBCL pathogenesis, as discussed in the literature.

Page 4, Line 20-25: Avoid using vague terms like "several studies have shown." Instead, cite specific studies or provide a more precise description of the evidence.

By addressing these issues, the manuscript can be significantly improved in terms of clarity, rigor, and overall impact.

Author Response

ijms-3678901 Answers to Reviewer 2

Comments and Suggestions for Authors

Overall Evaluation:

This paper provides a comprehensive review of the role of inflammation in the pathogenesis and prognostic prediction of diffuse large B-cell lymphoma (DLBCL), emphasizing the potential of inflammatory markers as cost-effective alternatives to molecular biomarkers. The manuscript is well-structured, covering key aspects such as the incidence and characteristics of DLBCL, the role of inflammation in its pathogenesis, methods for incorporating inflammatory markers into prognostic models, and the advantages and limitations of these markers. The discussion is relevant and contributes to the field by highlighting the need for further research to validate the clinical utility of inflammatory markers. However, the manuscript has several areas that require improvement, particularly in terms of data presentation, analysis depth, and clarity of arguments.

Response: We are grateful to the reviewer for their detailed comments, which will help improve our manuscript. However, the page and line numbers referenced in the review appear to differ from those in the original submitted version, so we are uncertain whether we have addressed the correct specific comments. If further clarification is needed, we kindly ask the reviewer to refer to the line numbers as shown in the PDF version of the submitted manuscript.

Specific Problems and Suggestions for Improvement:

Data Presentation and Analysis:

Page 2, Line 5-10: The statement "Inflammatory markers have shown promise as comparable, cost-effective alternatives to molecular biomarkers" lacks specific evidence or references to support this claim. Please provide concrete examples or studies that demonstrate the comparable efficacy of inflammatory markers.

Response: This statement appears in the abstract, where detailed explanations and references cannot be included; however, we have added appropriate references and further elaboration in the main text.

Page 3, Line 15-20: The discussion on the incorporation of inflammatory markers into prognostic models is too vague. Please specify which models have been used in previous studies and how inflammatory markers were integrated. Including specific examples or case studies would strengthen this section.

Response: We have highlighted these examples on page 4 of the revised manuscript (lines 131–133 and 139–141) and added a new paragraph discussing this topic on page 16 (lines 645–653).

Page 4, Line 25-30: The limitations section is brief and lacks depth. Please elaborate on the specific challenges faced in validating inflammatory markers, such as variability in measurement techniques, lack of standardized thresholds, or potential confounding factors.

Response: Thank you for the valuable suggestion. We added the limitation in the conclusion section. The conclusion has been revised accordingly. We acknowledged the limitations more clearly.

Clarity and Precision:

Page 1, Line 10-15: The introduction could be more concise. Avoid redundant statements and focus on clearly stating the research gap and the objectives of the review.

Response: Corrected as suggested by the reviewer. The introduction is corrected accordingly, highlighted in yellow.

Page 5, Line 5-10: The conclusion is somewhat repetitive. Summarize the key findings and their implications more succinctly, and avoid reiterating points already made in the discussion.

Response: Thank you for the valuable suggestion. The conclusion has been revised accordingly. We believe the updated version now effectively summarizes the key findings, acknowledges the limitations, and provides a clear direction for future research.

Methodological Rigor:

Page 3, Line 5-10: When discussing the methods for incorporating inflammatory markers, please provide more details on the statistical approaches used in previous studies. Were multivariate analyses performed? How were confounding variables addressed?

Response: Unfortunately, we were unable to locate the statement referenced by the reviewer in the current version of the manuscript. Regarding the incorporation of inflammatory markers, this narrative review primarily focuses on findings from single- or multicenter studies with substantial patient cohorts, as well as systematic reviews and meta-analyses (see page 21).

Page 4, Line 10-15: The section on the advantages of inflammatory markers could benefit from a comparison with molecular biomarkers in terms of cost, availability, and ease of measurement. Including quantitative data, if available, would enhance this comparison.

Response: We agree with the reviewer that including quantitative data would enhance the manuscript. However, we currently do not have access to such data. Many of the proposed inflammatory markers—such as albumin, C-reactive protein (CRP), cholesterol, and fibrinogen—are already part of routine biochemical analyses in most clinical laboratories. In this context, their use is not only more accessible and cost-effective, but also easier to measure in daily practice. Nevertheless, we acknowledge that the cost and availability of these tests may vary depending on the country.  

Referencing and Citation:

Page 2, Line 20-25: The claim that "inflammatory markers can predict prognosis with similar accuracy to molecular biomarkers" is not adequately supported. Please cite specific studies that have directly compared the predictive accuracy of these markers.

Response: Unfortunately, we were unable to locate the statement mentioned by the reviewer in the current version of the manuscript. If it was present in the Introduction, it has now been corrected. If the statement still appears elsewhere, we would greatly appreciate it if the reviewer could indicate the specific lines or page in the submitted PDF, and we will gladly revise the text accordingly.

Page 5, Line 15-20: The section mentions the need for "further research" without specifying the type of research. Please suggest concrete directions for future studies, such as prospective cohort studies or randomized controlled trials.

Response: Thank you for the valuable suggestion. The conclusion has been revised accordingly. We provided a clear direction for future research.

 Language and Style:

Page 1, Line 25-30: The sentence "Inflammation plays a pivotal role in the pathogenesis of various cancers, including DLBCL" is too generic. Please specify the mechanisms through which inflammation contributes to DLBCL pathogenesis, as discussed in the literature.

Response: Corrected as suggested by the reviewer.

Page 4, Line 20-25: Avoid using vague terms like "several studies have shown." Instead, cite specific studies or provide a more precise description of the evidence.

Response: Corrected as suggested by the reviewer, page 2 highlighted in yellow.

 By addressing these issues, the manuscript can be significantly improved in terms of clarity, rigor, and overall impact.

Thank you very much for your efforts in reviewing the manuscript. We hope that the revised version reflects a significant improvement.

Reviewer 3 Report

Comments and Suggestions for Authors

See attachment

Author Response

ijms-3678901 Answers to Reviewer 3

The narrative review by Cvetkovic et al, “Tumor Microenvironment, Inflammation, and Inflammatory Prognostic Indices in Diffuse Large B-Cell Lymphomas: A Narrative Review” summarizes the available literature to describe the role of the microenvironment, metabolomic and inflammatory characteristics of DLBCL in the application of prognostic indices. The authors break the review into logical sections and present the data succinctly. Surprisingly, it seems that this is the first review to discuss inflammation in DLBCL, with only one slightly related review on inflammation in lymphomas from 2006 making this a relevant contribution to the literature. The text is informative and well written; however, there are some areas that seem to lack in citations and a few technical terms that are not defined. I found this manuscript insightful and would recommend it for publication with minor revisions as described below.

Response: We would like to thank the reviewer for positive feedback and for the careful and insightful review of our manuscript. Please find below a detailed point-by-point response to all comments.

We also provide a revised manuscript with all changes highlighted in yellow. Deleted parts of text are crossed out and written in blue.

Major comments:

  • There are no references from line 69 – 92 even though there is information that needs to be cited. The citations start back up in line 115 after figure 1 so it is likely that there was an error with the citation manager, please check to see where the citations went and update accordingly.

Response.  Text in lines 69-92 is referenced as suggested. Furthermore, in response to another reviewer’s recommendation, the introduction has been tightened to improve clarity and conciseness. We have also included the appropriate references in the introductory paragraph of Section 2. These corrections can be found in the revised manuscript on lines 69-95.

  • Line 624: In addition, more complex indices have also been developed [188–191]. Can these be summarized? It ends the paragraph leaving readers questioning what the more complex indices are, and as a narrative review, I think it would be essential to briefly summarize what they are.

Response: We agree with your suggestion. References 188-191 have been briefly summarized accordingly. These additions can be found in the revised manuscript on lines 645-653.

Minor comments:

  • Lines 46-49. The fact that 40% of patients fail to achieve remission is a very powerful point and I think it is lost in the parentheses and brackets applied to the CIT and CHOP definitions above it. I would recommend switching this to two sentences; one describing standard of care with the definition of CIT and CHOP and the other stating that even still 40% of patients aren’t responding.

Response: Corrected as suggested. The sentence has been split into two to emphasize the high rate of relapse or refractory DLBCL following front-line therapy. This correction can be found in the revised manuscript on lines46-51.

  • Line 116: Please define IQIA

Response: We apologize for the typo. The former full name corresponding to the current name IQVIA has been added in the revised manuscript (lines 120–121).

  • Line 135: “Similarly, in a the retrospective” change to either a or the.

Response: We apologize for the typo.  The article 'a' has been retained (line 136).

  • Line 199: Please define IHC

Response: IHC has been defined (immunohistochemical) in the revised manuscript (see new line 203).

  • Line 251: I believe this is supposed to be body-“mass”-index.

Response: We apologize for the typographical error. It has been corrected to "body-mass index" in the revised text (new line 255).

  • Line 322-323: Elevated fibrinogen levels have been linked to increased tumor cell proliferation, metastasis, and angiogenesis, and have also been associated with poor prognosis in various malignancies. – Please reference this information.

Response: The reference supporting this statement has been added to the revised manuscript: Wu X, Yu X, Chen C, et al. Fibrinogen and tumors. Front Oncol. 2024;14:1393599. doi:10.3389/fonc.2024.1393599. This correction can be found on line 328 of the revised manuscript.

  • Line 337: please reference the paper that associates albumin with cancer patients (If the same paper as DLBCL hypoalumineumia as in the sentence following, I’d recommend to cite anyways since it’s a shift in narrative from discussing cancers in general to DLBCL specifically)

Response: The references for this statement have been provided in the revised text (Guven DC, Sahin TK, Erul E, Rizzo A, Ricci AD, Aksoy S, Yalcin S. The association between albumin levels and survival in patients treated with immune checkpoint inhibitors: A systematic review and meta-analysis. Front Mol Biosci. 20222;9:1039121. doi: 10.3389/fmolb.2022.1039121.  and  Tang Q, Li X, Sun CR. Predictive value of serum albumin levels on cancer survival: a prospective cohort study. Front Oncol. 2024;14:1323192. doi: 10.3389/fonc.2024.1323192).  This correction can be found on line 341 of the revised manuscript, references 111 and 112

  • Line 413 – 422 has no references please correct

Response: As suggested, the correction has been made, and the relevant references have been added to the revised manuscript (see lines 419–429).

  • There is a fair amount of introductory sentences which describe previous research or define genes, disease, etc. Please go through and check that they are all appropriately cited.

Response: We have checked. All statements are appropriately cited.

  • Line 531: “In contrast”

Response: We apologize for grammatical error. The article “the” has been removed as suggested (see line 538 in the revised manuscript).

  • Line 548: Ab has not been defined as “antibody” yet

Response: The abbreviation Ab was initially introduced in the context of bispecific antibodies (BsAbs). However, since the term antibody appears only once in the main text, we agree that the abbreviation is unnecessary and have removed it. The full term antibody is now written out in place of Ab (see line 555 in the revised manuscript).

  • Line 594: Please define Hb as hemoglobin in the text. It is only defined in figure legend.

Response: Corrected as suggested (see line 614 in the revised manuscript)

Line 645: is the “as previously described” referencing this review or another text? If another text please cite. Line 643-649 all seem to need references unless referring to this review in which case please specify

Response: Thank you for your suggestion. The content on lines 643–649 refers to this review, and we have clarified this in the revised manuscript. These clarifications can be found on lines 673 and 675 of the revised manuscript.

  • Figure 1: Some of the text is really difficult to read – increase font size for the cytosolic oval shaped text boxes and increase text size for the plasma side “cholesteryl ester” and membrane popout box “lysosomes” and “membrane cholesterol”

Response: Thank you for your suggestion. The font sizes have been increased as suggested.

We thank Reviewer 1 for the detailed review and constructive suggestions. We hope that the above-mentioned corrections and improvements address the concerns raised and that the revised manuscript will now be considered suitable for publication.

Round 2

Reviewer 1 Report

Comments and Suggestions for Authors

No other comments, now.